# SEQUENTIAL BRICK ASSEMBLY WITH EFFICIENT CONSTRAINT SATISFACTION

## ABSTRACT

We tackle the problem of assembling LEGO bricks, which can be considered as an instance of combinatorial optimization problems. Such a problem is challenging, since the number of possible structures increases exponentially with the number of available bricks due to complex physical constraints between bricks. To solve this problem, our method assesses a brick structure to predict the next brick position and its confidence by employing a U-shaped sparse 3D convolutional network. A convolution filter efficiently validates the physical constraints in a parallelizable and scalable manner and effectively allows us to process different brick types. To generate a novel structure, we devise a sampling strategy to determine the next brick position by considering attachable positions under the physical constraints. Instead of using handcrafted brick assembly datasets, our model is trained with a large number of 3D objects that enable to create a new high-fidelity structure. We demonstrate that our method successfully generates diverse brick structures while handling two different brick types and outperforms existing methods based on Bayesian optimization, deep graph generative model, and reinforcement learning, all of which consider assembling a single brick type. To show the validity of our method, elaborate studies on various assembly scenarios are also presented.

## 1 INTRODUCTION

Most real-world 3D structures are constructed with smaller *primitives*. A broad range of studies have tackled an interesting assembly problem such as molecule generation (Ertl et al., 2017; Neil et al., 2018; You et al., 2018), building construction (Talton et al., 2011; Martinovic & Van Gool, 2013; Ritchie et al., 2015), and part assembly generation (Sung et al., 2017; Lee et al., 2021; Jones et al., 2021). However, if a primitive unit (e.g., LEGO bricks) is used to construct a 3D structure, it is thought of as an instance of combinatorial optimization problems. In particular, the search space of the primitive assembly problem increases exponentially as a search depth increases: given $n$ primitives with $k$ possible combinations, the search space increases as $\mathcal{O}(n^k)$. In addition, supposing that a constraint between unit primitives should be satisfied, the problem becomes more challenging.

In this work, we tackle the problem of assembling LEGO bricks into a 3D structure. The generation problem of brick assembly is defined as a sequential decision-making process that sequentially appends a new brick by determining the brick type, position, and direction of the new brick. Moreover, a brick to assemble has to satisfy the physical constraints related to the disallowance of overlap, no isolated bricks, and LEGO-like brick connections; see Section 2 for the details of such constraints. Compared to general 3D generation methods such as previous work (Wu et al., 2016; Gadelha et al., 2018; Achlioptas et al., 2018), the advantages of sequential brick assembly as a generation method are that it can generate a structure in an open space and provide instructions to build the structure.

Several attempts have been conducted on the generation of sequential brick assembly by utilizing Bayesian optimization (Kim et al., 2020), deep graph generative model (Thompson et al., 2020), and reinforcement learning (Chung et al., 2021). Those methods are capable of assembling $2 \times 4$ LEGO bricks with consideration of the physical constraints. However, the previous literature has several limitations. A Bayesian optimization method proposed by Kim et al. (2020) requires heavy computations due to their iterative optimization process for each brick position. Thompson et al. (2020) propose to use masks to filter out invalid actions along with their method, but the utilization of masks degrades assembly performance. To predict a valid action, Chung et al. (2021) propose

Table 1: Comparisons to the previous approaches and our method in terms of unit primitives, algorithms, target conditioning, and constraint satisfaction. SA stands for sequential assembly. Note that our approach does not provide guidance (images or target shapes) when generating a new shape.

| Method | Unit Primitives | Algorithm | Target Conditioning | Constraint Satisfaction |
|--------|-----------------|-----------|---------------------|------------------------|
| SA with BayesOpt (Kim et al., 2020) | Single type | Bayesian optimization | Target volume | Subsampling |
| DGMLG (Thompson et al., 2020) | Single type | Graph generative model | Class label | Masking |
| Brick-by-Brick (Chung et al., 2021) | Single type | Reinforcement learning | Single- or multi-view images | Supervised learning |
| BrECS (Ours) | Two types | Supervised learning | Incomplete target volume (only for the completion task) | Convolution operations |

to employ a neural network, which fails to predict proper moves perfectly. More importantly, these methods share common limitations: they only consider using a single brick type (i.e., 2×4 brick) due to their inherent design choices or technical difficulties, and inevitably become slower in validating the constraints as the number of bricks increases due to exponentially increasing search spaces.

To tackle the limitations above, we devise a novel brick assembly method with a U-shaped sparse 3D convolutional neural network utilizing a convolution filter to validate complex brick constraints efficiently. By a sampling procedure proposed in this work, our method can create diverse assembly sequences of structures in a training stage, which makes us create training episodes. Also, we use the sampling procedure to generate high-fidelity brick structures. In addition, a convolution filter to validate the physical constraints enables our method to be easily parallelizable and scalable to processing different brick types. Henceforth, we refer to our method as sequential **Br**ick assembly with **E**fficient **C**onstraint **S**atisfaction (BrECS).

We carry out two scenarios of sequential brick assembly: completion and generation tasks. A completion task is to assemble bricks from a given partial structure, and a generation task is to create a brick structure from scratch; see Section 5 for their details. In the completion task, BrECS achieves 30.5% higher IoU scores than the best performing baseline on average (as reported in Table 2), while ours is eight times faster than the baseline. Also, in the generation task, BrECS performs the best with 117% higher average classification scores than the next best one (as reported in Table 3).

In summary, the contributions of this work are as follows:

- We propose a novel generation model for sequential brick assembly, which validates physical constraints with a convolution filter in parallel and generates high-fidelity 3D structures;
- We propose a U-shaped 3D sparse convolutional network that can be trained with a voxel dataset, which enables us to assemble high-fidelity structures by exploiting a popular 3D dataset, i.e., ModelNet40 (Wu et al., 2015);
- We show that our model successfully assembles two different brick types, i.e., 2×4 and 2×2 LEGO bricks, in various experiments.

We will release our codes and datasets upon publication.

## 2 RELATED WORK

**3D Shape Generation.** For 3D shape generation, point cloud and voxel representation are widely used. Various methods have been proposed to generate a point cloud, including methods using variational auto-encoders (Gadelha et al., 2018), generative adversarial networks (Achlioptas et al., 2018), and normalizing flows (Yang et al., 2019). On the other hand, voxel-based 3D shape generation has been studied. Wu et al. (2016) propose to use a generative adversarial network to generate voxel occupancy. A method, which is suggested by Choy et al. (2019), generates 3D shapes on voxel grids with a fully convolutional neural network that includes efficient sparse convolution. Zhang et al. (2021) propose a method that generates high-quality voxel-based shapes by iteratively applying 3D convolutional neural networks. A point cloud means that points on the cloud are continuous, borderless, and (possibly) infinite. In contrast, voxels are similar to pixels – coordinates are discretized and limited. Our method employs a voxel-based generation model to generate brick-wise scores since candidates for brick positions are discrete, and the number of bricks is finite.

**Sequential Part Assembly.** Similar to the sequential brick problem, a sequential part assembly problem also shares common difficulties to the combinatorial optimization problem, i.e., exponentially increasing search spaces. Ghasemipour et al. (2022) aim to assemble multi-part objects and tackle the problem with large-scale reinforcement learning and graph-based model architecture. Sung et al. (2017) assemble incomplete 3D parts with a part retrieval network and a position prediction network. The retrieval model is trained with an embedding network, so corresponding parts have similar representations in a low-dimensional feature space. Hu et al. (2020) propose a model that sequentially moves a piled box into another container with different shapes. Their approach has to predict box removal orders and box positions in a new container. They tackled the problem by representing previous boxes into graphs and reinforcement learning with rewards considering physical constraints in the new container. In contrast to these methods, we tackle the sequential *brick* assembly problem, which involves identical bricks to create 3D structures.

**Sequential Brick Assembly.** By following previous literature (Kim et al., 2020; Thompson et al., 2020; Chung et al., 2021), we sequentially assemble given LEGO bricks into a 3D object by adding on bricks one by one. Unlike generic 3D object generation methods, e.g., previous literature (Wu et al., 2016; Achlioptas et al., 2018), these sequential assembly approaches have to consider particular physical constraints that are introduced by available connections between two adjacent LEGO bricks and the disallowance of brick overlap. As discussed in the work (Kim et al., 2020), these constraints encourage us to accentuate the nature of combinatorial optimization since there exist a huge number of possible combinations in the presence of complex physical constraints.

## 3  SEQUENTIAL BRICK ASSEMBLY PROBLEM

**Formulation.** Similar to the reference by Chung et al. (2021), a new brick has to be connected to one or (possibly) more bricks of previously assembled bricks. It implies that the position of the next brick can be determined as a relative distance from a certain brick directly connected to the next brick. We denote this indicator brick as a *pivot brick*. Eventually, after choosing the pivot brick, we determine a relative position from the pivot brick so we can place the new brick on the position determined. To choose pivot brick and relative position effectively, we must consider physical constraints and scores for every possible brick position. As the number of bricks increases, determining the pivot brick and the relative position becomes challenging because of exponentially increasing successive brick positions and complex physical constraints. Consequently, previous methods struggle to generate fine structures, and they tend to use a smaller number of bricks. Furthermore, while the recent work (Kim et al., 2020; Thompson et al., 2020; Chung et al., 2021) only focuses on sequential assembly with $2 \times 4$ bricks, we tackle $2 \times 4$ and $2 \times 2$ bricks with a more efficient framework. The summary of each approach is shown in Table 1.

**Constraints.** We should take into account three physical constraints in the brick assembly problem: (i) bricks should not overlap each other (NO-OVERLAP); (ii) all bricks of the current structure should be connected to each other, and no isolated bricks exist (NO-ISOLATION); (iii) a new brick must be directly attached to the upper or lower position of other bricks (VERTICAL ASSEMBLE). However, validating such constraints is not trivial. To overcome these difficulties, as presented in Table 1, Kim et al. (2020) sample a subset of available next brick positions, Thompson et al. (2020) mask out invalid positions by validating all possible positions, and Chung et al. (2021) train an auxiliary network that is trained to validate positions.

## 4  PROPOSED APPROACH

We explain four steps of our method, which are illustrated in Figure 1. To sum up, we generate a brick structure under physical brick constraints by repeating four steps of the next brick position score computation (Section 4.1.1), invalid position exclusion (Section 4.1.2), pivot brick sampling (Section 4.1.3), and relative brick position determination (Section 4.1.4). The neural network for next brick position score computation (utilized in Section 4.1.1) is the only learnable component of our framework. The details of the training procedure and loss function for the proposed network are described in Section 4.2. The details of the procedure are presented in Algorithms 1 and 2.

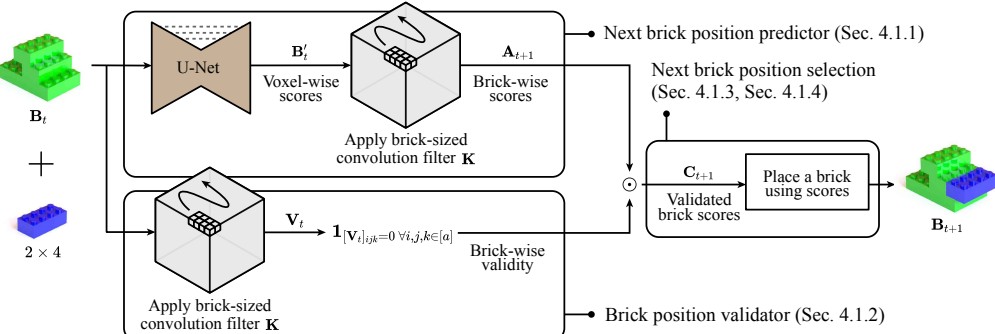

Figure 1: Illustration of the proposed efficient constraint satisfaction method with convolution filters for sequential brick assembly. U-Net and a symbol $\odot$ denote a U-shaped sparse 3D convolutional neural network and element-wise multiplication, respectively.

## 4.1 EFFICIENT CONSTRAINT SATISFACTION

In this section, we propose a novel method to tackle the challenge of satisfying the following physical constraints: NO-OVERLAP, which is validated by using convolution operations; NO-ISOLATION and VERTICAL ASSEMBLE, which are satisfied by following the brick assembly formulation with pivot bricks and relative brick positions, presented in Chung et al. (2021). By borrowing the concept of constraint satisfaction (Tsang, 1993), which is the problem of finding solutions that satisfy a predefined set of constraints, our method is designed to establish an approach to efficient constraint satisfaction for sequential brick assembly.

### 4.1.1 PREDICTING NEXT BRICK POSITIONS

Given a voxel representation of structure at step $t$, which is denoted as $\mathbf{B}_t \in \{0, 1\}^{a \times a \times a}$ where $a$ is the size of 3D space, we first feed the voxel representation $\mathbf{B}_t$ into a U-shaped sparse 3D convolutional neural network, inspired by the reference (Choy et al., 2019): $\mathbf{B}'_t = \text{U-Net}(\mathbf{B}_t)$, in order to capture global and local contexts effectively and retain the same dimensionality. Due to its pyramidal feature extraction structure, the U-Net extracts robust features understanding multi-dimensional contexts. We validate that the U-Net effectively extracts important contexts and thus improves overall performance compared to other neural networks in Table 5. Moreover, we expect that our neural network produces a likely complete or potentially next-step 3D structure, which is represented by a probability of voxel occupancy. Note that the network parameters in this U-Net are only the learnable component in our framework BrECS. The U-Net gives guidance on natural shapes. In this section, we assume that we are given pretrained U-Net, and the training procedure is described in Section 4.2.

After obtaining $\mathbf{B}'_t \in \mathbb{R}^{a \times a \times a}$, a score for next brick positions $\mathbf{A}_{t+1} \in \mathbb{R}^{a \times a \times a}$ is computed by sliding a convolution filter $\mathbf{K} \in \mathbb{R}^{w_b \times d_b \times 1}$ across $\mathbf{B}'_t$: $\mathbf{A}_{t+1} = \mathbf{B}'_t * \mathbf{K}$, where $*$ is a convolution operation. Note that we match the size of $\mathbf{A}_{t+1}$ to the size of $\mathbf{B}_t$ by applying zero padding. In particular, the size of the convolution filter is the same as the brick size we assemble $w_b \times d_b$ where $w_b$ and $d_b$ are the width and depth of the brick, respectively so that we can determine the scores over all the possible positions of the next brick by aggregating the corresponding voxels where the convolution filter is applied. For example, if we use $2 \times 4$ bricks, the size of the convolution filter is $2 \times 4 \times 1$. Moreover, $\mathbf{K}$ is always initialized as a tensor filled with 1 without updating its values in a training stage to aggregate $w_b d_b$ voxels equally with a single convolution operation. In the case of assembling a structure with $r$ brick types, we use $2r$ convolution filters of the same size with $r$ brick types and repeat the above process for each convolution filter. In particular, for each brick type, we employ two convolutional filters (i.e., $w_b \times d_b \times 1$ and $d_b \times w_b \times 1$) to consider brick rotation.

### 4.1.2 FILTERING OUT INVALID BRICK POSITIONS

As shown in Figure 1, to predict the validity of the next brick positions, a brick-sized convolution filter is applied to $\mathbf{B}_t$: $\mathbf{V}_t = \mathbf{B}_t * \mathbf{K}$. The filter $\mathbf{K}$ is identical to the filter used in computing the score for the next brick positions. After applying the convolution filter across $\mathbf{B}_t$, all possible brick

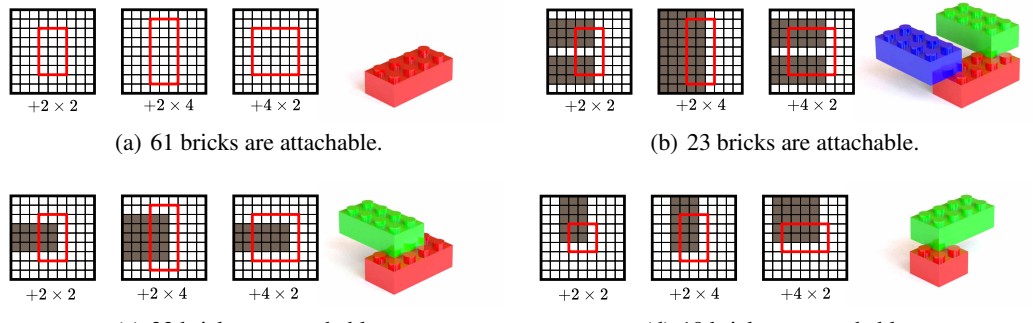

(a) 61 bricks are attachable.

(b) 23 bricks are attachable.

(c) 33 bricks are attachable.

(d) 19 bricks are attachable.

Figure 2: Illustration of brick structures and their possible brick positions. The three grids on the left of the bricks show the validity $\mathbf{V}_{t+1}$ of each brick position on the bottommost red brick for brick type $2\times2$, $2\times4$, and rotated $2\times4$. The pixels in a red rectangle indicate attachable brick positions on the bottommost red brick without considering overlap. The gray pixels indicate that overlap will occur if a new brick is attached to the corresponding position.

positions are determined if the value of the position of interest is zero, which means that no overlap exists within the brick positions. As visually presented in Figure 2, the validity of each position in terms of overlap and the number of attachable positions with respect to the brick of interest are readily determined by applying a convolution filter of the corresponding brick type.

Using these two branches: computing $\mathbf{A}_{t+1}$ with the U-shaped network and $\mathbf{V}_t$, we can define a validated score for next brick positions $\mathbf{C}_{t+1}$. Each entry of $\mathbf{C}_{t+1}$ is equal to its score for the next brick positions filtering out invalid brick positions due to overlaps. Formally, we describe $\mathbf{C}_{t+1}$ of assembling a $w_b \times d_b$ brick on a previously assembled brick of size $w_p \times d_p$ as follows:

$$\mathbf{C}_{t+1} = \mathbf{1}_{[\mathbf{V}_t]_{ijk}=0 \ \forall i,j,k\in[a]}(\mathbf{V}_t) \odot \mathbf{A}_{t+1}$$
$$= \mathbf{1}_{[\mathbf{B}_t*\mathbf{K}]_{ijk}=0 \ \forall i,j,k\in[a]}(\mathbf{B}_t * \mathbf{K}) \odot (\text{U-Net}(\mathbf{B}_t) * \mathbf{K}), \quad (1)$$

where $\mathbf{1}_{[\mathbf{V}_t]_{ijk}=0 \ \forall i,j,k\in[a]}$ is an indicator function, $\mathbf{K} \in \mathbb{R}^{w_b \times d_b \times 1}$ is the same convolution filter as in Section 4.1.1, and $*$ is a convolution operation. Then, we assemble a structure by iteratively placing a brick on one of the positions with non-zero scores; the details of how the position is determined are described in Section 4.1.3. Since the calculation of the masked score tensor $\mathbf{C}_{t+1}$ uses convolution operations, we efficiently compute it with modern GPU devices. Specifically, the validity check of $(64, 64, 64)$ voxels without parallelization takes 4.7 seconds. On the other hand, the validity check of $(64, 64, 64)$ voxels with our method takes only 0.0021 seconds, which implies that this design serves *2,270 times faster* inference on the validity check. In the next subsection, we describe the detailed sampling procedure for the next brick positions.

### 4.1.3 SELECTING PIVOT BRICKS BY SAMPLING

To place a single brick, we need to choose one of the previously assembled bricks to attach the new brick to make the entire brick assembly connected – as mentioned in Section 2, we refer to it as a pivot brick. The motivation of our method to select a pivot brick is that a pivot brick with higher sum of attachable brick scores should be more preferable than one with lower sum of attachable brick scores, rather than choosing a pivot brick that is connected to a position with the highest score of $\mathbf{C}_{t+1}$. Since a neural network tends to memorize training samples and their assembly sequences, choosing a position with the highest score fails to create novel structures. To avoid a deterministic approach based on $\mathbf{C}_{t+1}$, we alter a method to select a pivot brick into a probabilistic method by sampling. To compare the number of attachable positions, we define a pivot score $T_{ijk}$ of the pivot of $(i, j, k)$ to aggregate scores of attachable positions:

$$T_{ijk} = \sum_{l=i-(\lfloor w' \rfloor - 1)}^{i+(\lceil w' \rceil - 1)} \sum_{m=j-(\lfloor d' \rfloor - 1)}^{j+(\lceil d' \rceil - 1)} \sum_{n \in \{k-1, k+1\}} \mathbf{C}_{lmn}, \quad (2)$$

where $w' = \frac{w_b + w_p}{2}$, $d' = \frac{d_b + d_p}{2}$, and $w_p \times d_p$ is the size of a previously assembled brick. After computing pivot scores, we can determine the most probable pivot brick using $\arg\max$ operation.

However, as mentioned above, we employ a sampling strategy to determine a pivot brick:

$$(a, b, c) \sim \mathbf{p}, \quad \text{where} \quad [\mathbf{p}]_{ijk} = p_{ijk} \left( \frac{T_{ijk}}{\sum_{(l,m,n) \in \text{Pivots}} T_{lmn}} \right). \tag{3}$$

Our sequential procedure for pivot brick predictions is inspired by Monte-Carlo tree search (MCTS) (Coulom, 2006). MCTS evaluates possible actions by expanding a search tree with Monte-Carlo simulations and back-ups. Similar to this, our method also evaluates pivot brick candidates by aggregating their attachable brick scores, which can be understood as a search tree with depth 1. Our method utilizes the U-shaped sparse 3D convolutional network to predict possible brick positions, which is analogous to a policy network of the recent MCTS method (Silver et al., 2016), which employs a neural network to predict prior distributions for search tree expansion.

### 4.1.4 DETERMINING RELATIVE BRICK POSITIONS

After choosing a pivot brick by following a method described in Section 4.1.3, a relative brick position to the pivot brick is determined to complete the brick placement. Possible relative brick positions $(x, y, z)$ of assembling a $w_b \times d_b$ brick on a previously assembled brick of size $w_p \times d_p$ are inherently integer-valued positions satisfying following conditions:

$$\left\lfloor \frac{w_b + w_p}{2} \right\rfloor - 1 \leq x \leq \left\lceil \frac{w_b + w_p}{2} \right\rceil - 1, \quad \left\lfloor \frac{d_b + d_p}{2} \right\rfloor - 1 \leq y \leq \left\lceil \frac{d_b + d_p}{2} \right\rceil - 1, \quad z \in \{-1, 1\}. \tag{4}$$

By considering the conditions described in Equation 4, we choose the relative brick position $(x, y, z)$ with the highest score of $\mathbf{C}_{t+1}$. A brick must be attached to the pivot brick as we add scores of every attachable bricks in Equation 2. In this step, we do not employ a sampling based method due to its poor empirical assembly results.

### 4.2 TRAINING THE SCORE FUNCTION

Similar to language modeling (Mikolov et al., 2010; Sutskever et al., 2014) and reinforcement learning (Sutton & Barto, 2018), our model also predicts a next brick position sequentially, which makes a problem challenging. To train such a prediction model, a pair of ground-truth state transition is required as a training sample. However, a final voxel occupancy is only available as ground-truth information. It implies that we cannot access intermediate states explicitly. By replacing $\mathbf{B}'_t$, i.e., the output of the U-Net, with a ground-truth voxel occupancy, we generate an assembly sequence $[\widetilde{\mathbf{B}}_0, \widetilde{\mathbf{B}}_1, \ldots, \widetilde{\mathbf{B}}_{T-1}]$ following the procedure in this section. Generated sequences are *unique and diverse*, since the stochasticity is injected from the sampling strategy introduced in Section 4.1.3.

Since there exist numerous possible sequences to assemble bricks to a certain 3D structure, a single-step look-ahead with a pair of contiguous states, i.e., $(\widetilde{\mathbf{B}}_t, \widetilde{\mathbf{B}}_{t+1})$, is not enough to model practical assembly scenarios. In addition, the training becomes unstable even though the training pairs are slightly changed. To address these issues, we train our sequential model to predict a $k$-step look-ahead state using pairs of states at step $t$ and step $t + k$, i.e., $(\tilde{\mathbf{B}}_t, \tilde{\mathbf{B}}_{t+k})$. From now, we call this technique *sequence skipping*.

We use a voxel-wise binary cross-entropy to train our model. By restricting the voxel prediction using a sigmoid function and minimizing the voxel-wise binary cross-entropy, our model learns to predict valid voxel-wise probabilities of the Bernoulli distribution. To sum up, we train our sequential prediction model as follows:

1. We generate a sequence of brick assembly $[\widetilde{\mathbf{B}}_0, \widetilde{\mathbf{B}}_1, \ldots, \widetilde{\mathbf{B}}_T]$ by running our brick assembly method with a ground-truth voxel occupancy in a training dataset;

2. We generate multi-step pairs from $[\widetilde{\mathbf{B}}_0, \widetilde{\mathbf{B}}_1, \ldots, \widetilde{\mathbf{B}}_T]$ in a sliding-window fashion, i.e., $\{(\tilde{\mathbf{B}}_t, \tilde{\mathbf{B}}_{t+k})\}_{t=0}^{T-k}$;

3. We train a transition function $p_\theta(\tilde{\mathbf{B}}_{t+k} | \tilde{\mathbf{B}}_t)$ with $\{(\tilde{\mathbf{B}}_t, \tilde{\mathbf{B}}_{t+k})\}_{t=0}^{T-k}$ and the voxel-wise binary cross-entropy.

Table 2: Quantitative results of the completion of brick structures. An asterisk after a method name denotes that partial or full ground-truth information is given to the corresponding model.

| Methods | IoU ($\uparrow$) | | | | % valid ($\uparrow$) | | | | Inference Time (sec., $\downarrow$) | | | |
|---|---|---|---|---|---|---|---|---|---|---|---|---|
| | airplane | table | chair | avg. | airplane | table | chair | avg. | airplane | table | chair | avg. |
| BayesOpt* | 0.145 | 0.206 | 0.233 | 0.194 | **100.0** | **100.0** | **100.0** | **100.0** | 1.20e6 | 1.11e6 | 1.05e6 | 1.12e6 |
| Brick-By-Brick* | 0.455 | 0.440 | 0.434 | 0.443 | 12.0 | 7.0 | 16.0 | 11.7 | 305.6 | 1502.4 | 2785.2 | 1531.1 |
| DGMLG | 0.315 | 0.269 | 0.271 | 0.285 | 0.0 | 1.0 | 0.0 | 0.3 | 237.3 | 340.0 | 473.0 | 350.4 |
| BrECS ($2 \times 4$) | 0.571 | 0.586 | 0.534 | 0.564 | **100.0** | **100.0** | **100.0** | **100.0** | **36.3** | **143.9** | **151.0** | **110.4** |
| BrECS ($2 \times 4 + 2 \times 2$) | **0.599** | **0.594** | **0.541** | **0.578** | **100.0** | **100.0** | **100.0** | **100.0** | 73.8 | 224.1 | 279.0 | 192.3 |

To decorrelate the time steps of training pairs in a batch and shorten training time, we employ a buffer throughout training to store training pairs similar to Zhang et al. (2021). The detailed training process is described in Algorithm 2 of the appendix.

**Inference for Sequential Brick Assembly.** In an inference stage, we determine the next brick positions by following our aforementioned procedure, given $\mathbf{B}_0$. We provide a different form of $\mathbf{B}_0$ depending on a task, i.e., completion and generation, and sequentially generate $\mathbf{B}_t$ until a terminal step $T$. For a completion task, we use an intermediate state, i.e., incomplete brick structure, as $\mathbf{B}_0$. For a generation task, we sample an initial brick position from a discrete uniform distribution, i.e., $(x, y, z) \sim \mathcal{U}(\{-2, 2\}^3)$. Then, we assemble bricks on a zero-centered voxel grid of size $(64, 64, 64)$ and use the voxel occupancy of the initial brick position sampled as $\mathbf{B}_0$.

## 5 EXPERIMENTAL RESULTS

We demonstrate that our model generates diverse structures with high fidelity satisfying physical brick constraints in the experiments on the completion of brick structures (see Section 5.1), and the experiments on the generation of brick structures with distinct brick types, e.g., $2 \times 4$ and $2 \times 2$ LEGO bricks (see Section 5.2). Also, we thoroughly validate each component of BrECS in ablation studies.

**Dataset.** To generate ground-truth assembly sequences and the training pairs based on the ground-truth sequences, we use the ModelNet40 dataset (Wu et al., 2015). 3D meshes in the dataset are converted into $(64, 64, 64)$-sized voxel grids, and then they are scaled down to 1/4 of the original size to reduce the number of required bricks. Other details are described in the supplement.

**Metric.** For the completion task, we use intersection over union (IoU) to evaluate the performance:

$$\text{IoU}(\mathbf{B}^{(1)}, \mathbf{B}^{(2)}) = \frac{\sum_{i=1}^{a} \sum_{j=1}^{a} \sum_{k=1}^{a} [\mathbf{B}^{(1)}]_{ijk} \cap [\mathbf{B}^{(2)}]_{ijk}}{\sum_{i=1}^{a} \sum_{j=1}^{a} \sum_{k=1}^{a} [\mathbf{B}^{(1)}]_{ijk} \cup [\mathbf{B}^{(2)}]_{ijk}}, \quad (5)$$

where $\mathbf{B}^{(1)}, \mathbf{B}^{(2)} \in \mathbb{R}^{a \times a \times a}$. Along with IoU, we measure the ratio of valid bricks in structures of interest. In addition, we utilize a class probability of a target class, which is the softmax output of the target class, in experiments on the generation of brick structures. The probability of the target class is measured using a pretrained classifier with the ModelNet40 dataset reported.

**Baseline Methods.** We compare the brick assembly performance of our method against Bayesian optimization (Kim et al., 2020), denoted as BayesOpt, Brick-by-Brick (Chung et al., 2021), denoted as BBB, and deep generative model of LEGO graphs (Thompson et al., 2020), denoted as DGMLG, in Tables 2 and 3. BayesOpt optimizes brick positions to maximize intersection over union (IoU) between assembled shapes and target shapes. BBB learns to assemble bricks given multi-view images of target structures using reinforcement learning. Following its formulation, we also provide three images (top, left, and front) of target structures in a test dataset. DGMLG generates a structure by utilizing the graph representation of brick structures and a deep graph generative model. Note that our approach does not provide any guidance (image or target shape) to produce a new structure.

### 5.1 COMPLETION OF BRICK STRUCTURES

We conduct our method on a completion task for a sequential brick assembly where unseen partial structures are given. To measure the completion performance on brick assembly problems, we first

Table 3: Quantitative results of the generation of brick structures with distinct brick types. Asterisk denotes that partial or full ground-truth information is given to the corresponding model.

| Methods | Class probability of target class (↑) | | | | % valid (↑) | | | |
|---|---|---|---|---|---|---|---|---|
| | airplane | table | chair | avg. | airplane | table | chair | avg. |
| BayesOpt* | 0.039 | 0.043 | 0.069 | 0.050 | **100.0** | **100.0** | **100.0** | **100.0** |
| Brick-By-Brick* | 0.430 | 0.042 | 0.032 | 0.168 | 6.0 | 3.0 | 2.0 | 3.7 |
| DGMLG | 0.228 | 0.023 | 0.027 | 0.093 | 0.0 | 0.0 | 0.0 | 0.0 |
| BrECS ($2 \times 4$) | 0.415 | **0.250** | 0.404 | 0.356 | **100.0** | **100.0** | **100.0** | **100.0** |
| BrECS ($2 \times 4 + 2 \times 2$) | **0.447** | 0.229 | **0.419** | **0.365** | **100.0** | **100.0** | **100.0** | **100.0** |

Figure 3: Qualitative results of brick assembly generation. Each row corresponds to specific baselines. Our method generates detailed brick assemblies without brick overlap. Best viewed in color.

assemble LEGO bricks using voxel occupancy in a test dataset. Then, we remove a fraction of bricks assembled without losing connectivity between bricks and provide it as a start state $\mathbf{B}_0$. Each model is trained with a training dataset and then complete brick structures from the start states. We compare the completion performance by measuring IoU between ground-truth voxel occupancy and the complete brick structure. In addition, we report valid assembly ratio and inference time.

As shown in Table 2, our method outperforms the other three baseline methods in terms of IoU. The results show that our method assembles high-fidelity brick structures compared to other methods, despite exhaustive constraint satisfaction. Moreover, our method performs the best in valid ratio alongside BayesOpt, but ours is also the best in inference time. We additionally test our model with distinct brick types by appending $2 \times 2$ bricks after assembling $2 \times 4$ brick type first. The performance of our method is further improved by using two brick types since different brick types can fill brick positions more densely and express the fine aspect of structure in consequence.

## 5.2 GENERATION OF BRICK STRUCTURES

As our method is a generation model, our brick assembly model can generate a brick structure that belongs to a particular category in a training dataset. To compare the quality of generated structures semantically, we train a classifier over voxel grids with a small number of 3D convolution layers using the ModelNet40 dataset; the detailed architecture of the classifier is presented in the appendix. Given a pretrained classifier over voxel grids, we measure the class probability of generated brick structures for a target class. To feed the voxel grid of generated structure into the classifier, we match the voxel grid size of generated structure with the size of the training dataset for the classifier.

Table 4: Results of ablation study on completion from intermediate states of chair category in the ModelNet40 dataset. Our model without ablation performs better than ablated variants. The performance is further improved with the additional $2 \times 2$ brick type.

|  | IoU | % valid | # of bricks | Inference time (sec.) |
|---|---|---|---|---|
| BrECS ($2 \times 4$; default) | 0.534 | 100.0 | 96.7 | 151.0 |
| BrECS ($2 \times 4 + 2 \times 2$) | 0.541 | 100.0 | 125.9 | 279.0 |
| BrECS (w/o validity check) | 0.441 | 0.0 | 141.6 | 335.8 |
| BrECS (w/o sequence skip) | 0.437 | 100.0 | 59.3 | 6.0 |

The quantitative results are presented in Table 3. Our method achieves the highest score in terms of class probabilities. The results indicate that our method generates a semantically meaningful brick structure compared to other methods. We also emphasize that the semantic generation quality can be improved using additional $2 \times 2$ brick types. Brick structures with distinct brick types are generated by assembling $2 \times 4$ bricks first and then $2 \times 2$ LEGO bricks. This performance gain is led by the additional improvement on structure refinement that cannot be filled with a single brick type.

## 5.3 ABLATION STUDY

We verify that each of our components in BrECS improves generation performance in the ablation study. Specifically, we compare the ablated variants of our method in terms of completion performance. Similar to the previous completion experiments, we complete a brick structure from the intermediate state of the unseen structure in a test dataset. The results of the ablation study are presented in Table 4. Following the previous experiments, creating brick structures with two brick types improves the performance. A validity check using a convolution filter plays a critical role in constraint satisfaction. Interestingly, the validity check also improves generation performance, i.e., IoU, since we train the model with valid assembly sequences. Moreover, our model performance degrades significantly without sequence skipping as the number of bricks added reduces. It supports our claim in Section 4.2 that the sequence skipping encourages our model BrECSto predict diverse brick positions. More detailed ablation studies can be found in the appendices.

## 6 CONCLUSION

We proposed a brick assembly method that efficiently generates high-fidelity brick structures satisfying physical constraints. Our model sequentially assembles bricks by repeating pivot selection and relative position selection. To aggregate brick scores and validate brick positions, we employ a convolution filter. Previous brick assembly methods cannot escape from exponentially increasing search spaces since it is the nature of combinatorial optimization problems. However, our method is efficient where the number of bricks increases as convolution operations are easily parallelizable on GPU devices. In addition, we show that our method performs better than prior methods, as shown in quantitative and qualitative results. To our best knowledge, we are the first to attempt to use multiple brick types, i.e., $2 \times 4$ and $2 \times 2$ bricks, to generate novel brick structures sequentially.

**Potential Applications.** Our model can be thought of as an instance of combinatorial optimization problems, which is a challenging problem in computer science. Moreover, our model can be adopted to solve a real-world problem such as building design and product assembly with an industrial robot. Considering such problems, we would like to emphasize that they suffer from coping with invalid actions or physical constraints. Thus, ours can help to solve such problems by utilizing our ideas.

**Limitations.** We utilize only two types of bricks for the demonstration, but we have designed that our model can use any rectangular bricks without an explicit definition of connection rules between different brick types, unlike the work (Kim et al., 2020). Hence, we can increase the number of brick types for sequential brick assembly. Future work will employ more diverse brick types and even irregular brick types since non-rectangular materials are common in real-world assembly scenarios. Another limitation is that our performance degrades when we use two brick types simultaneously. In the case of using $2 \times 4$ and $2 \times 2$ brick types, smaller bricks are used more frequently compared to a scenario with only $2 \times 4$ brick type. We presume that smaller $2 \times 2$ bricks in-between reduce attachable positions, resulting in overall performance degradation. Also, we would like to consider physical stability, which is essential for real-world applications.

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

APPENDIX

In the appendix, we cover the detailed descriptions missing in the main manuscript.

## A    IMPLEMENTATION DETAILS OF OUR METHOD AND BASELINE MODELS

### A.1    BRECS

To train 3D voxel generation efficiently, we utilize MinkowskiEngine (Choy et al. (2019)) and its 3D sparse convolution operation. We train our model with fixed learning rate of 5e-4, Adam optimizer (Kingma & Ba, 2015), batch size of 32, sequence skipping with step size $k = 8$, buffer size of 1024, and maximum number of bricks of 150. The input size of our model is $(64, 64, 64)$ and output size is $(64, 64, 64)$. We trained the model until reaching 100k steps.

### A.2    BAYESIAN OPTIMIZATION

We employ Bayesian optimization to tackle a sequential brick assembly problem. We follow the setup proposed by Kim et al. (2020). The number of bricks are limited to 160 bricks at most. Every brick position is optimized by maximizing IoU between currently assembled and ground-truth structures. We would like to emphasize that we have to provide ground-truth voxel information to this strategy.

### A.3    BRICK-BY-BRICK

We use Brick-by-Brick as a baseline method and compare brick assembly performances. We follow the model architecture and training setup of the model described in (Chung et al., 2021). We also would like to emphasize that we have to provide partial ground-truth voxel information to this strategy, as the method requires 3 images of ground-truth shape to assemble target shapes. The number of bricks are limited to 75 bricks at most, which is the same as original setup, due to the excessively increasing memory requirements of the model. In inference, the sequence of generation is halted when the newly placed brick violates brick constraints, following the environment reset condition of the method.

### A.4    DEEP GENERATIVE MODEL OF LEGO GRAPHS

We solve brick assembly generation problem using Deep Generative Model of LEGO Graphs and compare the performance against ours. We follow the model architecture and training setup of the method described in (Thompson et al., 2020). To train the graph generation model, we need graphs of target shapes. To this end, we assemble target shapes with ground truth voxel shapes. Then we convert the brick assemblies into graphs by representing bricks as nodes and direct connections between bricks as edges, following (Thompson et al., 2020). We train the model for 200 epochs.

## B    BRICK ASSEMBLY PROCESS

We describe brick assembly process in Algorithm 1.

## C    TRAINING

We describe brick assembly model training procedure in Algorithm 2.

## D    DETAILED ARCHITECTURE OF U-NET

We design our score prediction network inspired by Choy et al. (2019). Detailed model architecture is illustrated in Figure 4.

---

**Algorithm 1** Brick Assembly Process

---

**Input:** Voxel of structure at current time step $\mathbf{B}$, assembled brick position list $\mathcal{P}$ and brick size $w_b \times d_b$

**Output:** Pivot brick position $(a, b, c)$, relative next brick position $(x, y, z)$

    Initialize convolution filter $\mathbf{K} \in \mathbb{R}^{w_b \times d_b \times 1}$ with 1, pivot score list $\mathcal{T} = \phi$, and possible relative connections $\mathcal{N}$ with values satisfying the conditions of Equation 4

    Calculate $\mathbf{C}$ using Equation 1                                          ▷ Section 4.1.2

    **for** each pivot $(a, b, c)$ in $\mathcal{P}$ **do**

        Calculate $T_{abc}$ using Equation 2

        $\mathcal{T} \leftarrow \mathcal{T} \cup \{T_{abc}\}$

    **end for**

    Sample pivot $(a, b, c) \sim \frac{T_{abc}}{\sum_{T \in \mathcal{T}} T}$                            ▷ Section 4.1.3

    Relative next brick position $(x, y, z) = \arg\max_{(a,b,c) \in \mathcal{N}} \mathbf{C}_{a+x, b+y, c+z}$     ▷ Section 4.1.4

---

**Algorithm 2** Brick Assembly Model Training

---

**Input:** Dataset $\mathcal{D}$, mini-batch size $M$, sequence skipping value $k$

**Output:** Brick assembly model parameter $\theta$

    Generate ground-truth brick assembly sequences $\widetilde{\mathbf{B}}_{0:T}^{(i)}$ and store in $\mathcal{D}_s$ using voxel shapes $x_i \in \mathcal{D}$

    Initialize buffer $\mathcal{B}$ with $(\widetilde{\mathbf{B}}_{0:T}^{(j)}, x_j, 0)$ where $x_j \sim \mathcal{D}, \widetilde{\mathbf{B}}_{0:T}^{(j)} \in \mathcal{D}_s$

    **repeat**

        $\mathcal{L} = 0$

        Sample and remove mini-batch $\{(\widetilde{\mathbf{B}}_{0:T}^{(i)}, x_i, t_i)\}_{i=1}^{M}$ from $\mathcal{B}$

        **for** $(\widetilde{\mathbf{B}}_{0:T}^{(i)}, x_i, t_i)$ in a mini-batch **do**

            $\mathcal{L} \leftarrow \mathcal{L} + \log p_\theta(\widetilde{\mathbf{B}}_{t_i+k}^{(i)} | \widetilde{\mathbf{B}}_{t_i}^{(i)})$

            **if** $t_i + k = T$ **then**

                Push $(\widetilde{\mathbf{B}}_{0:T}^{(j)}, x_j, 0)$ into $\mathcal{B}$ where $x_j \sim \mathcal{D}, \widetilde{\mathbf{B}}_{0:T}^{(j)} \in \mathcal{D}_s$

            **else**

                $\mathcal{B} \leftarrow \mathcal{B} \cup \{(\widetilde{\mathbf{B}}_{0:T}^{(i)}, x_i, t_i + 1)\}$

            **end if**

        **end for**

        $\theta \leftarrow \theta + \eta \frac{\partial \mathcal{L}}{\partial \theta}$

    **until** convergence

---

## E   Detailed Architecture of Voxel Classifier

We train a shallow voxel classification network to compare assembly generation quality semantically. We add softmax layer at the end of the network to make output logit to be a distribution. Detailed network structure is described in Figure 5.

## F   Ablation Study on Model Architectures

We compare different model architectures for a score prediction model. Specifically, we compare U-Net and fully convolutional networks. For a fair comparison, we use the same number of convolutional filters. We present the results of generation and completion on Table 5. We find that the U-Net performs better than fully convolutional network in both generation and completion. This is due to the robust feature extraction from pyramidal structure of U-Net.

## G   Model Performance without Stochasticity

We carry out ablation study on the effects of stochasticity in a brick position selection process and compare completion and generation performance against the original model. The quantitative results are presented in Table 6. Stochasticity improves generation performance as expected. Interestingly, a completion quality is also improved by stochasticity. In the completion task, we remove half of

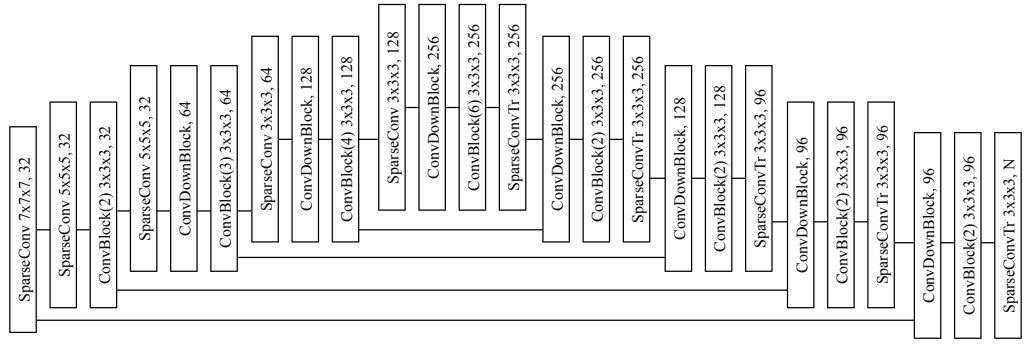

Figure 4: Detailed model architecture of U-Net.

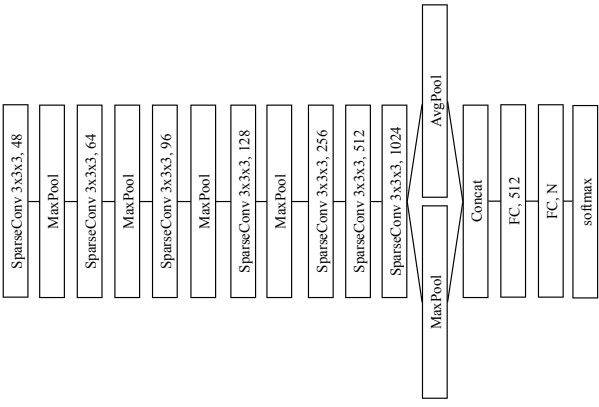

Figure 5: Detailed model architecture of voxel classifier.

the bricks and the model has to complete partial assembly results, which lose the majority of their information. Stochasticity may help to infer its original shape by making multiple candidates. Wan et al. (2021) also report that stochasticity improves FID score of image completion task if they are masked out large area of original images.

## H    PREVENTION OF SQUEEZING-IN BRICKS

A brick position can be predicted in the middle of bricks. We call this brick as a squeezing-in brick. A Squeezing-in brick is easily prevented with simple modification to our pipeline. As illustrated in Figure 6, we employed additional $1 \times 1 \times 3$ convolutional filters after we obtain brick-wise validity. Then we check whether the resultant validity score is 2. The value 2 means the brick position and one of neighbor brick positions (top or bottom) is empty.

## I    ANALYSIS ON DIVERSITY OF GENERATED SHAPES

We measure precision and recall to measure the diversity of generated shapes.

We define precision as how much of real shape distribution can be made using a generated shape distribution and recall as how much of generated shape distribution can be made using a real shape distribution by following the work by Sajjadi et al. (2018). Distance between distributions is measured in the feature space encoded by a pretrained voxel classifier. The results are depicted in Figure 7. We find that the diversity is different for each class. Airplane class lacks diversity as the class have

Table 5: Results of ablation study on model architectures. U-Net architecture consistently outperforms fully convolutional network with the same number of convolutional filters in both reconstruction and generation scenarios.

| Methods | IoU (↑) | | | | Class probability of target class (↑) | | | |
|---|---|---|---|---|---|---|---|---|
| | airplane | table | chair | avg. | airplane | table | chair | avg. |
| BrECS (U-Net) | 0.571 | 0.586 | 0.534 | 0.564 | 0.415 | 0.250 | 0.404 | 0.356 |
| BrECS (Fully convolutional network) | 0.510 | 0.399 | 0.450 | 0.453 | 0.208 | 0.075 | 0.186 | 0.156 |

Table 6: Model performance without stochasticity. Stochasticity improves both generation performance and completion performance.

| Methods | IoU (↑) | | | | Class probability of target class (↑) | | | |
|---|---|---|---|---|---|---|---|---|
| | airplane | table | chair | avg. | airplane | table | chair | avg. |
| BrECS (U-Net) | 0.571 | 0.586 | 0.534 | 0.564 | 0.415 | 0.250 | 0.404 | 0.356 |
| BrECS (w/o stochasticity) | 0.515 | 0.533 | 0.509 | 0.519 | 0.238 | 0.104 | 0.217 | 0.186 |

high precision but low recall. Table class shows balanced results between diversity and similarity. Chair class has diverse shapes as it has high recall but low precision.

## J    BRICK ASSEMBLY CONSIDERING INTERMEDIATE STABILITY

Humans assemble bricks in a way that every intermediate structure is stable as depicted in Figure 8. Thus, we conduct brick assembly generation considering intermediate stability. To compare intermediate stability, we introduce a new evaluation metric, called a trajectory stability ratio. The metric measures how many intermediate structures are stable by dividing the number of stable structures by the total number of structures. We consider that a structure is stable when the center point of the structure does not move within given simulation steps. We utilize PyBullet to simulate stability of generated shapes. The quantitative results are presented in Table 7.

We measure stability and generation performance of our method and its variants. The variants are the models that (i) start assembling from bottom and (ii) consider intermediate stability. To consider intermediate stability during inference, we exclude unstable actions by simulating stability of possible outcomes. We find that the variant considering intermediate stability shows higher trajectory stability ratio against other variants. This scenario is only conducted in sequential brick assembly generation and one step generation method is not appropriate for this case.

## K    ADDITIONAL QUALITATIVE RESULTS OF OUR MODEL

Additional qualitative results of our model is shown in Figure 9.

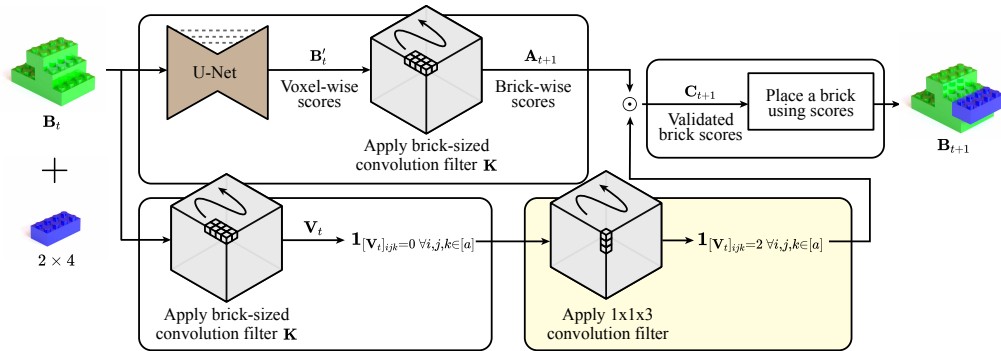

Figure 6: Illustration of the proposed method with squeezing-in brick prevention. The additional part is emphasized in yellow.

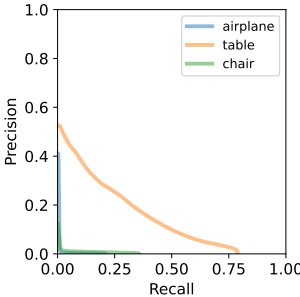

Figure 7: Result of precision and recall for generated shapes. Distance between distributions is measured on the feature space encoded by a pretrained voxel classifier.

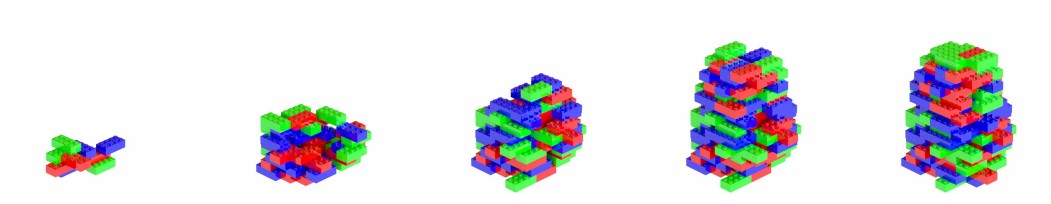

Figure 8: An assembly sequence where every intermediate structure is stable.

Table 7: Quantitative result of shape generation and trajectory stability. Changed initial brick position improves stability and stability check further improves stability.

| Methods | Class probability of target class (↑) | | | | trajectory stability ratio (↑) | | | |
|---|---|---|---|---|---|---|---|---|
| | cup | lamp | vase | avg. | cup | lamp | vase | avg. |
| BrECS (Start from center) | 0.032 | 0.084 | 0.102 | 0.073 | 0.543 | 0.737 | 0.62 | 0.633 |
| BrECS (Start from bottom) | 0.046 | 0.088 | 0.074 | 0.069 | 0.892 | 0.772 | 0.846 | 0.837 |
| BrECS (w/ intermediate stability) | 0.024 | 0.022 | 0.056 | 0.034 | 0.925 | 0.921 | 0.936 | 0.927 |

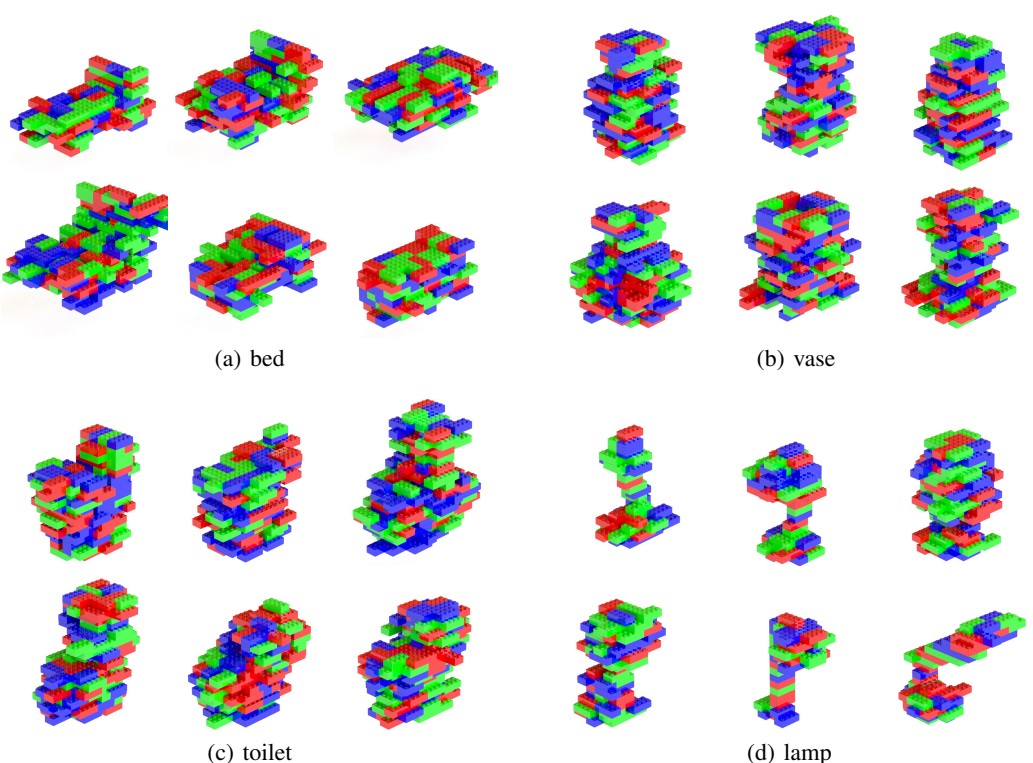

(a) bed

(b) vase

(c) toilet

(d) lamp

Figure 9: Additional qualitative results of our model.

