# OpenReview forum: "Sequential Brick Assembly with Efficient Constraint Satisfaction"
_ICLR.cc/2023/Conference — Submitted to ICLR 2023_

### Official Review · Reviewer_LQmJ · 2022-10-23

**Confidence:** 4
**Correctness:** 2
**Technical Novelty And Significance:** 2
**Empirical Novelty And Significance:** 3
**Recommendation:** 3

**Clarity, Quality, Novelty And Reproducibility:**

- the flow is unclear in places – e.g. sec. 4.1 says it looks at satisfying constraints, yet 4.1.1 instead addresses generating unconstrained 'possible next step' voxelisations

- aside from this the writing is generally good – clear and fluent, with very few typos

- the precise tasks addressed are not defined very clearly early in the paper – inputs, outputs, etc.

- the task is novel, as is the overall approach proposed. Within the pipeline, all the components are fairly standard

- it appears that sufficient details are given to allow reimplementation of the proposed method (particularly if the authors make the code public as promised)


**Strength And Weaknesses:**

- the first task considered is generation of models of certain object classes, from Lego. However, this is straightforwardly accomplished by using an existing generative model of ModelNet shapes (outputting any representation that can be converted to voxels), then applying Brickr [Testuz, Eurographics 2013] to convert this to a valid Lego model. That approach already supports many more brick types than the proposed method, and supports all the relevant constraints.

- the second task considered is completing partial lego models of ModelNet classes. This can again be addressed easily using existing techniques – convert to a point cloud, run a point-cloud completion model, convert back to voxels, and run Brickr on the 'new' part, giving the uppermost part of the input model as the first Brickr layer.

- very few qualitative generation examples are given (three examples from two classes) – making it impossible to judge whether generated shapes are in fact reasonable. Those which are given, are not particularly impressive, and it's not specified whether they are curated.

- there is no quantification (based on nearest-neighbors) of whether the method is in fact generating novel shapes, or simply memorising its training set

- the constraints in sec. 3 need more explanation. Do they correctly disallow 'squeezing in' blocks? Why must structure be fully-connected from start (submodules assembled separately then combined are common in real Lego)?

- the overall approach is a complex engineered pipeline, very specific to the task, and most stages of this are (in my opinion) not of great interest to the majority of the ICLR community

- stochasticity in the generative process appears to be introduced only by randomly selecting which of a candidate set of points to consider at each step. This is fine a-priori; however, the set of candidates appears to be determined not by the learnt model directly, but by the number of possible locations that bricks could be affixed to them. Why not let the model learn the distribution over 'next' attachment points? Also, why is the selection of 'pivot brick' artificially separated from choosing which particular location on this brick to attach to?


**Summary Of The Paper:**

Authors propose a method for sequential generation of Lego models. This addresses two tasks – (i) generating new class-conditional Lego models a priori; (ii) completing partial models. The proposed approach consists of a neural network to predict how good candidate locations for the 'next' brick is, and various checks to ensure constraints on fabricability are fulfilled. The method is demonstrated on shapes from the ModelNet dataset, and shown to perform favorably versus the chosen baselines.

**Summary Of The Review:**

As noted above, I believe the tasks can be addressed straightforwardly using existing methods (at very least, these should be included as a baseline). Moreover, the pipeline is not particularly interesting technically, nor do there seem to be many generalisable insights that would be beneficial to the community.

---

> ### Author Response · Authors · 2022-11-19
> **Responses to Reviewer LQmJ**
>
> Thank you for your valuable comments. Here we answer the questions and concerns described in your review.
> ***
> > Generation of models of certain object classes is straightforwardly accomplished by using an existing generative model of ModelNet shapes, then applying Brickr [Testuz, Eurographics 2013] to convert this to a valid Lego model. That approach already supports many more brick types than the proposed method, and supports all the relevant constraints.
>
> > Completing partial lego models can again be addressed easily using existing techniques - convert to a point cloud, run a point-cloud completion model, convert back to voxels, and run Brickr on the 'new' part, giving the uppermost part of the input model as the first Brickr layer.
>
> Of course you can create a brick structure using a voxel generator. However, there are some advantages of sequential shape generation. First, our model assembles unit bricks to a structure, always satisfying the physical constraints. In contrast, other voxel generator-based brick assembly approaches require inefficient constraint verification steps by converting voxels into 1x1 bricks and merging them into larger bricks recursively. Second, a sequential brick generation approach predicts a sequence of brick structure. As real-world construction considers a sequential assembly, our model better fits to the real-world scenario and easily transferred to such cases.
> ***
> > Very few qualitative generation examples are given (three examples from two classes)
>
> We presented more additional qualitative results of our model; see ***Appendix K*** and ***Figure 9***.
> ***
> > there is no quantification (based on nearest-neighbors) of whether the method is in fact generating novel shapes, or simply memorising its training set
>
> We analyzed the diversity of generated shapes from our model using a precision-recall graph; see ***Appendix I*** and ***Figure 7***. The graph shows that the diversity depends on the classes of shapes. The classes with balanced or high recall show that our model can generate diverse shapes, not memorizing the original distribution of the training dataset.
> ***
> > the constraints in sec. 3 need more explanation. Do they correctly disallow 'squeezing in' blocks?
>
> It is a good point. We allowed 'squeezing in' blocks. In the revision we added the experiments on this issue by slightly modifying our model architecture. Now, you can find the results that disallow 'squeezing in' blocks; see ***Appendix H*** and ***Figure 6***.
> ***
> > Why must structure be fully-connected from start (submodules assembled separately then combined are common in real Lego)?
>
> As described in Table 1, we followed the previous literature to discuss those methods and compare them. Moreover, by this interesting constraint that only allows full connectivity, we make this problem more combinatorial.
> ***
> > the overall approach is a complex engineered pipeline, very specific to the task, and most stages of this are (in my opinion) not of great interest to the majority of the ICLR community
>
> We disagree with this comment. As opposed to your comment "it is a complex engineered pipeline", we effectively utilize a convolution filter of brick size to consider physical constraints.
> We think that it can be a simple, but effective method. By using the convolution filter, we also enjoy the benefits of modern GPU-enabled softwares including PyTorch and TensorFlow.
> ***
> > Why not let the model learn the distribution over 'next' attachment points? Also, why is the selection of 'pivot brick' artificially separated from choosing which particular location on this brick to attach to?
>
> We empirically found that learning next attachable points degrades performance. This is due to the additional burdens of excluding a current structure and inferring possible connected positions compared to predicting the next shape only. In addition, we should choose pivot bricks to make every brick connected to each other. We computed the summation of the scores of next brick positions to obtain a pivot brick score. We clarified motivation of our method to select a pivot brick; see ***section 4.1.3***.
> ***
> > the flow is unclear in places – e.g. sec. 4.1 says it looks at satisfying constraints, yet 4.1.1 instead addresses generating unconstrained 'possible next step' voxelisations
>
> To explain a method to validate whether constraints are satisfied, we need to present Section 4.1.1 first. We did not change the flow in the revision because we think that the current flow is reasonable, but we will clarify more in the final version.
> ***
> > the precise tasks addressed are not defined very clearly early in the paper – inputs, outputs, etc.
>
> By considering your comment, we revised our submission; please see ***Section 1***.
> ***
> > it appears that sufficient details are given to allow reimplementation of the proposed method (particularly if the authors make the code public as promised)
>
> We promise that we will make our implementation publicly available.

---

### Official Review · Reviewer_khxJ · 2022-10-23

**Confidence:** 3
**Correctness:** 3
**Technical Novelty And Significance:** 3
**Empirical Novelty And Significance:** 3
**Recommendation:** 5

**Clarity, Quality, Novelty And Reproducibility:**

- Clarity is not enough in my opinion for the reasons mentioned above.
- The problem formulation and technical approaches are novel compared to previous works.
- Reproducibility is not an issue for me if the authors are willing to release the code.

**Strength And Weaknesses:**

Strengths:
- Technically sound methodology. The technical flow in Figure 1 is intuitive.
- Good empirical performance in terms of shape completion accuracy, semantic generation quality, validity, and efficiency compared to baselines.

Weaknesses:
- The paper is poorly written with confusing logic, and the task definition is unclear until the end of the paper. Such poor clarity is unacceptable for more potential readers. See more details below.
- The wider impact or applications of the research problem are not clear. Though the authors believe this method can be extended to building design or industrial robots, the proposed method seems highly specific to LEGO bricks and hard to extend.

Paper writing and clarity issues:
- The abstract does not illustrate which problem/goal specifically you are targeting. Obviously, you are working on brick sequence generation, but is the generation conditioned on certain target shapes or shape classes? From the current abstract, it seems to be that you are generating ANY assembly sequence that satisfies the physical constraints, which confused me because I don't see why you need such a complicated approach to achieve that.
- The detailed definitions of completion and generation tasks are only vaguely described in Sec 5.1-5.2. In other words, most of the time when I was reading the paper, I didn't have a clear clue of which specific problem the authors were addressing. The terms "completion task" and "generation task" are frequently used in the middle of the paper without explanation. So please bring the task definitions forward - let people know what you are addressing even before describing the technical approach.

Questions:
- Sec 4.1.3 details the pivot brick selection procedure, where the authors propose sampling to avoid being deterministic. This is reasonable for the novel structure generation task. But for the shape completion task, does adding stochasticity help?
- In Sec 4.2, I am not sure if I understand this correctly: "By replacing $B_t^{'}$ with a ground-truth voxel occupancy, we generate an assembly sequence ...". Would you mind elaborating more on this procedure?

**Summary Of The Paper:**

This paper proposes a method that generates high-fidelity brick structures satisfying physical constraints for two tasks: partial structure completion and novel structure generation for a given class of shape. The main contributions are:
- A parallelizable brick position predictor and validator based on convolution filters, making the inference of the proposed brick sequence generation method efficient on GPU for large brick structures.
- The proposed method is agnostic to the brick type, with experimental demonstrations of assembling structures using both 2x4 and 2x2 bricks.

**Summary Of The Review:**

Though the results seem good, I cannot give acceptance based on the current state of the paper, as there is so much confusion. I look forward to the authors' clarification on important aspects.

---

> ### Author Response · Authors · 2022-11-19
> **Responses to Reviewer khxJ**
>
> Thank you for your valuable comments. Here we answer the questions and concerns described in your review.
> ***
> > The paper is poorly written with confusing logic, and the task definition is unclear until the end of the paper. Such poor clarity is unacceptable for more potential readers.
>
> According to the review by **Reviewer LQmJ**, "aside from this the writing is generally good - clear and fluent, with very few typos", we cannot fully agree with your comment. However, we revised our submission in order to clarify our method as well as our task. Please take a look at the revision.
> ***
> > The wider impact or applications of the research problem are not clear. Though the authors believe this method can be extended to building design or industrial robots, the proposed method seems highly specific to LEGO bricks and hard to extend.
>
> We revised the "Potential Applications" section; see ***Section 6***. In addition, we would like to emphasize that our goal is not only to solve a brick assembly task. We wish to generalize our idea to solving other combinatorial problems, e.g., building construction and molecule generation. Since such a task also suffers from considering physical constraints, we now focus on a more structural yet still interesting task.
> ***
> > The abstract does not illustrate which problem/goal specifically you are targeting. Obviously, you are working on brick sequence generation, but is the generation conditioned on certain target shapes or shape classes? From the current abstract, it seems to be that you are generating ANY assembly sequence that satisfies the physical constraints, which confused me because I don't see why you need such a complicated approach to
> achieve that.
>
> We revised the abstract and also the introduction section. Our method does not generate a structure conditioned on a certain target shape or a shape class. We generate it from scratch, which is similar to popular generative models including variational autoencoders and generative adversarial networks. Moreover, we tested more realistic and diverse results including stability consideration in the appendices; please see the revision.
> ***
> > The detailed definitions of completion and generation tasks are only vaguely described in Sec 5.1-5.2. In other words, most of the time when I was reading the paper, I didn't have a clear clue of which specific problem the authors were addressing. The terms "completion task" and "generation task" are frequently used in the middle of the paper without explanation. So please bring the task definitions forward - let people know what you are addressing even before describing the technical approach.
>
> We revised the introduction section by reflecting your comment. Thank you for pointing out this.
> ***
> > Sec 4.1.3 details the pivot brick selection procedure, where the authors propose sampling to avoid being deterministic. This is reasonable for the novel structure generation task. But for the shape completion task, does adding stochasticity help?
>
> In the completion task, we remove half of the bricks and the model has to complete the structure from partial information, which loses the majority of its information. In addition,
> stochasticity can help to infer its original shape by making multiple candidates. Wan et al. also report that stochasticity improves FID score of an image completion task if they mask out a large area of original images.
>
> To show similar results, we measured model performance without stochasticity to show that the effects of stochasticity in the next brick position selection improves model performance, especially in shape completion; see ***Appendix G*** and ***Table 6***.
>
> [Wan et al.] Wan Z. et al., High-Fidelity Pluralistic Image Completion with Transformers, ICCV, 2021.
> ***
> > In Sec 4.2, I am not sure if I understand this correctly: "By replacing with a ground-truth voxel occupancy, we generate an assembly sequence ...". Would you mind elaborating more on this procedure?
>
> By replacing $\mathbf{B}_t'$ with a ground-truth voxel occupancy for training, our model chooses a position that has higher overlap between the brick and ground-truth voxel occupancy.

---

### Official Review · Reviewer_sD14 · 2022-10-24

**Confidence:** 3
**Correctness:** 3
**Technical Novelty And Significance:** 3
**Empirical Novelty And Significance:** 3
**Recommendation:** 5

**Clarity, Quality, Novelty And Reproducibility:**

**Presentation**

    "Thompson et al. (2020) propose to use masks along with their
    method, but the utilization of masks degrades assembly
    performance."

What are masks?

---

    "these methods share common limitations: they only use a single
    brick type"

They *can't* be adapted to use multiple shapes or they were simply
tested with a single type?

    "inevitably become slower as the number of bricks increases due to
    an exponentially increasing search space."

I'm quite sure that this is true for the proposed method too. The
difference is that the proposed method can be easily parallelized on GPUs. If my understanding is correct, the computation for different candidate shapes can be carried out in parallel regardless of the assembly technique. In principle, the other techniques could be parallelized, right?

---

    "To tackle the limitations above, we devise a novel brick assembly method with a U-shaped sparse
    3D convolutional neural network utilizing a convolution filter to validate complex brick constraints
    efficiently."

What is the intuition though? I wouldn't describe in detail on how
the core idea is implemented ("U-shaped 3D CNN") in the first sentence
after the related work.

At the same time, the specific choice of neural architecture is never motivated in the rest of the paper.
Why using a U-Net? Have you considered (and performed ablation studies with) different architectures?

---

    "BrECS performs the best with 117% higher average classification scores than the next best one."

I have no idea what 117% higher average classification score
means. Does it mean "more than twice as accurate"? I wouldn't report
numerical results here. Even if the metric is well-known or clearly
explained, it is not really useful to the reader at this point.


---

    "Unlike the sequential brick problem, a sequential part assembly
    problem also shares similar difficulties to the combinatorial
    optimization problem [...]"

Unlike? Is 'sequential brick problem' different than 'sequential brick
ASSEMBLY problem'? If not, isn't the sequential brick assembly problem
a special (simpler) case of sequential part assembly problem?


---

    "these constraints encourage us to accentuate the nature of
    combinatorial optimization since there exist a huge number of
    possible combinations in the presence of complex physical
    constraints."

What do you mean with 'to accentuate the nature of CO'?

---

    "Moreover, we expect that our neural network produces a likely
    complete or potentially next-step 3D structure, which is
    represented by a probability of voxel occupancy."

It is not clear what Bt' should represent. An example would definitely
help. Is it a probability mass function over a x a x a? How do you
ensure that the U-net outputs a valid distribution?

---

    "In the case of assembling a structure with r brick types, we
    prepare r convolution filters of the same size with r brick types
    and repeat the above process for each convolution filter."

I guess that the actual number should be at most r * 4 (different
rotations for asymmetric shapes). Here you should briefly mention how
convolutions have to be adjusted for asymmetric shapes.

---

    "a score for next brick positions A(t+1) in R^(a×a×a) is computed by
    sliding a convolution filter"

An example here would also help. I don't get the intuition on why the
dimensionality of A(t+1) should be the same as Bt', isn't A(t+1)
scoring the different options in terms of brick positioning? These
should be less than R^(a×a×a).

The same doubts apply to V(t+1).

---

    "The motivation of our method to select a pivot brick is that a
    pivot brick with more attachable positions should be more
    preferable than one with less attachable positions, rather than
    choosing a pivot brick that is connected to a position with the
    highest score of C(t+1)."

Shouldn't the method actively try to maximise the objective?

    "Since a NN tends to memorize training samples and their assembly
    sequences, choosing a position with the highest score fails to
    create novel structures."

The fact that you don't expect your predictor to generalize doesn't
seem a good reason for disregarding the score.

Overall, I think that the sampling procedure is not clearly
presented. I would also expect a discussion on the
exploration-exploitation trade-off, i.e. how to control diversity
vs. quality of the generated structures.

---

    "By replacing Bt' , i.e., the output of the U-Net, with a
    ground-truth voxel occupancy, we generate an assembly sequence
    following the procedure of BrECS."

Isn't BrECS the proposed approach?


---

It is not clear which limitations are
inherent to the proposed approach and which ones can be addressed with
future work. Can the proposed approach accomodate any constraint
besides the ones considered here (e.g. constraints on the frequency of
certain blocks, or budgets)? My intuition is that filtering out infeasible configurations with convolutions only works for a restricted number of constraints, but I might be wrong.

---

**Experiments**

    "Generated sequences are unique and diverse, since the
    stochasticity is injected from the sampling strategy introduced in
    Section 4.1.3."

Isn't uniqueness implying diversity? More importantly, diversity is not measured in any way.

    "We demonstrate that our model generates diverse structures with
    high-fidelity satisfying physical brick constraints in the
    experiments on the completion of brick structures."

The trade-off between quality and diversity is not investigated in the
experiments.

High-fidelity is often mentioned in the paper but never formally
defined. After re-reading the paper I realized that the goal is to
generate structures that are similar to a reference model.
This aspect could be clarified.






**Strength And Weaknesses:**

Strengths:
- The proposed approach leverages tensorial computations with a significant performance gain
- The code will be released upon publication

Weaknesses (for details, see Clarity section below):
- The presentation could be significantly improved
- Some crucial aspects of the problem are not empirically validated

**Summary Of The Paper:**

This paper proposes an approach for the sequential brick assembly problem (SBA). To the best of my understanding SBA is a special case of sequential assembly problem with a finite number of actions (possible brick placements). The proposed approach is based on a combination of a convolutional NN, which is traned to predict the score of possible placements, and convolutions for quickly filtering out infeasible solutions with respect to given physical constraints.

**Summary Of The Review:**

While the proposed approach addresses some computational challenges of the problem considered, its presentation should be improved before publication. Little insights are provided on why the technique is effective and what are its limitations. The experimental section disregards important  aspects in this sequential generation, such as the exploration vs. explotation tradeoff.
All these aspects require a major revision of the paper. I cannot recommend the current version for publication.

---

> ### Author Response · Authors · 2022-11-19
> **Responses to Reviewer sD14**
>
> Thank you for your valuable comments. Here we answer the questions and concerns described in your review.
> ***
> > What are masks?
>
> > They can't be adapted to use multiple shapes or they were simply tested with a single type?
>
> We revised the explanation; see ***Sec. 1***. Thank you for pointing these out.
> ***
> > The computation for different candidate shapes can be carried out in parallel regardless of the assembly technique. In principle, the other techniques could be parallelized, right?
>
> For the brute-force constraint verification used in the previous work, they might be parallelized. However, our point is that a convolution filter makes our method much easier to use modern GPU-enabled softwares including PyTorch and TensorFlow.
> ***
> > I wouldn't describe in detail on how the core idea is implemented("U-shaped 3D CNN") in the first sentence after the related work. At the same time, the specific choice of neural architecture is never motivated in the rest of the paper. Why using a U-Net? Have you considered(and performed ablation studies with) different architectures?
>
> We utilize the U-Net in order to capture global and local contexts effectively and retain the same dimensionality. Due to its pyramidal feature extraction structure, the U-Net extracts robust features understanding multi-dimensional contexts. We conducted an ablation study on model architectures to show U-Net successfully extracts stable features and improves performance compared to other models; see ***Sec. 5***, ***Appendix F***, and ***Table 5***.
> ***
> > I have no idea what 117% higher average classification score means. Even if the metric is well-known or clearly explained, it is not really useful to the reader at this point.
>
> We clarified this part; please see the corresponding section.
> ***
> > Unlike? Is 'sequential brick problem' different than 'sequential brick ASSEMBLY problem'?
>
> We fixed this; we are sorry for the confusion.
> ***
> > What do you mean with 'to accentuate the nature of CO'?
>
> The nature of CO in the context means that the search space increases exponentially as a search depth increases.
> ***
> > It is not clear what Bt' should represent. How do you ensure that the U-net outputs a valid distribution?
>
> $\mathbf{B}_t'$ represents voxel-wise next shape probabilities of the Bernoulli distribution. By restricting the voxel prediction using a sigmoid function and minimizing the voxel-wise binary cross-entropy, our model learns to predict valid voxel-wise probabilities of the Bernoulli distribution; see ***Sec. 4.2***.
> ***
> > I guess that the actual number of required convolution filters should be at most 4  for different rotations for asymmetric shapes. Here you should briefly mention how convolutions have to be adjusted for asymmetric shapes.
>
> We restricted the brick shape to be rectangular and in this case, the number of different rotations is at most 2. Thus in the case of assembling a structure with $r$ brick types, we use $2r$ convolution filters of the same size with $r$ brick types; see ***Sec. 4.1.1***.
> ***
> > I don't get the intuition on why the dimensionality of A(t+1) should be the same as Bt', isn't A(t+1) scoring the different options in terms of brick positioning? The same doubts apply to V(t+1).
>
> As you mentioned, the dimensionality of A(t+1) does not have to be the same as Bt’. The dimensionality of A(t+1) has only to be the same as the dimensionality of Vt.
> ***
> > Shouldn't the method actively try to maximise the objective?
>
> We computed the summation of the scores of next brick positions to obtain a pivot brick score. We clarified the motivation of our method to select a pivot brick; see ***Sec. 4.1.3***.
> ***
> > The fact that you don't expect your predictor to generalize doesn't seem a good reason for disregarding the score.
>
> Our model does not disregard the score but rather respect the score. We sample a pivot using the normalized pivot brick score distribution, which contains scores from pivots with non-maximal score.
> ***
> > Overall, I think that the sampling procedure is not clearly presented. How to control diversity vs. quality of the generated structures?
>
> Currently, our model does not have any parameters or architectures to control the exploration-exploitation trade-off. However, we added the components to increase the diversity by sampling a pivot using a score distribution. We analyzed the diversity from our model using a precision-recall graph; see ***Appendix I*** and ***Fig. 7***.
> ***
> > ​​Isn't BrECS the proposed approach?
>
> We fixed this. Thank you for pointing this out.
> ***
> > Can the proposed approach accommodate any constraint besides the ones considered here(e.g. constraints on the frequency of certain blocks, or budgets)?
>
> Our method considers `no-overlap`, `no-isolation` and `vertical-assemble` constraints. However, we try to emulate the concept of budgets by using multiple brick types. We can reduce the number of bricks used by assembling with larger brick types and then assembling with smaller brick types.

---

### Official Review · Reviewer_3PGp · 2022-10-25

**Confidence:** 4
**Correctness:** 3
**Technical Novelty And Significance:** 3
**Empirical Novelty And Significance:** 3
**Recommendation:** 6

**Clarity, Quality, Novelty And Reproducibility:**

The paper is very clear and appears to be original. The method appears to be novel though the high-level problem has been addressed in other papers. The tech is non-trivial and plausible. For other quality assessments, please see my comments above.


**Strength And Weaknesses:**

The paper is well-written and presents a plausible and efficient solution to the problem. Long assembly sequences with a wide branching factor can be difficult to control, and the presented results look quite good considering this challenge. The method guarantees that the assemblies are valid by construction. The training is also self-supervised and does not appear to use explicit GT construction sequences.

That said, the (not great in absolute terms) quality of the results does leave one wondering if combining a voxel-based generator with a subsequent (non-ML) step that fits an arrangement of bricks to the voxel pattern would be a better way to generate (still structurally valid) assemblies of Lego bricks, rather than to impose distributional constraints on very long Markov sequences. The presented method is an interesting technical demo of controlling long assembly sequences with tight constraints, but its practical utility in the context of Lego construction seems limited. Autoregressive methods have their place, but in this case it seems like a particularly challenging paradigm to employ, and there don't seem to be obvious takeaways transferable to other sequential assembly problems. The "Potential Applications" section mentions building construction and robotic assembly, and in each of these tasks there seem to be easier ways to achieve the end goal.

Finally, even though the paper claims that the assembly sequences are ones humans can actually follow ("can... provide instructions to build the structure"), the sequences don't actually take things like phsyical stability and connection strength of intermediate structures into account, so this may not be true in all cases.


**Summary Of The Paper:**

This paper describes a method for sequentially assembling Lego bricks into 3D structures resembling objects from training classes. Efficient GPU-based methods are developed to filter out invalid attachment points and to learn to identify high-value placements. Unlike prior work, the method handles a heterogeneous mix of brick types (a mix of 2x4 and 2x2 bricks is demonstrated).

**Summary Of The Review:**

I think this paper does a good job in developing an improved algorithm for sequential Lego brick assembly. That said, there are some weaknesses which I have identified above. Hence, I think the paper, while interesting, does not represent a very significant step forward. If the authors could argue for greater impact of the algorithm, that assessment might change.

---

> ### Author Response · Authors · 2022-11-19
> **Responses to Reviewer 3PGp**
>
> Thank you for your valuable comments. Here we answer the questions and concerns described in your review.
> ***
>
> > That said, the (not great in absolute terms) quality of the results does leave one wondering if combining a voxel-based generator with a subsequent (non-ML) step that fits an arrangement of bricks to the voxel pattern would be a better way to generate (still structurally valid) assemblies of Lego bricks, rather than to impose distributional constraints on very long Markov sequences.
>
> Of course you can create a brick structure using a voxel generator. However, there are some advantages of sequential shape generation. First, our model assembles unit bricks to a structure, always satisfying the physical constraints. In contrast, other voxel generator-based brick assembly approaches require inefficient constraint verification steps by converting voxels into 1x1 bricks and merging them into larger bricks recursively. Second, a sequential brick generation approach predicts a sequence of brick structure. As real-world construction considers a sequential assembly, our model better fits to the real-world scenario and easily transferred to such cases.
> ***
>
> > Autoregressive methods have their place, but in this case it seems like a particularly challenging paradigm to employ, and there don't seem to be obvious takeaways transferable to other sequential assembly problems. The "Potential Applications" section mentions building construction and robotic assembly, and in each of these tasks there seem to be easier ways to achieve the end goal.
>
> We revised the "Potential Applications" section; see ***Section 6***. In addition, we would like to emphasize that our goal is not only to solve a brick assembly task. We wish to generalize our idea to solving other combinatorial problems, e.g., building construction and molecule generation. Since such a task also suffers from considering physical constraints, we now focus on a more structural yet still interesting task.
> ***
>
> > Finally, even though the paper claims that the assembly sequences are ones humans can actually follow ("can... provide instructions to build the structure"), the sequences don't actually take things like physical stability and connection strength of intermediate structures into account, so this may not be true in all cases.
>
> We tested a brick assembly scenario by considering intermediate stability using a physics simulator (PyBullet) in order to make the entire generation sequence stable; see ***Appendix J***, ***Figure 8***, and ***Table 7***. The results show that employing a physics simulator in the brick position selection increases the ratio of stable structures in generating a sequence. Currently, we consider intermediate stability during inference time.

---

### Author Response · Authors · 2022-11-19
**Overall Responses to All Reviewers**

We sincerely appreciate your valuable comments. By considering your reviews and carrying out additional experiments, we revised our submission. We describe a list of revisions as follows.

### Revisions

- Added the reason why previous researches use only single brick types (i.e., $2 \times 4$ bricks); see ***Section 1***.
- Explained why we employ U-Net to predict the next state brick positions; see ***Section 4.1.1***.
- Elaborated on the number of convolution filters we use when we consider brick rotation; see ***Section 4.1.1***.
- Clarified motivation of our method to select a pivot brick; see ***section 4.1.3***.
- Clarified the inference procedure of sequential brick assembly with our model and the task we tackle (i.e., completion task and generation task); see ***Section 4.2***.
- Stated that our approach does not provide any information to generate a new shape.
- Conducted an ablation study on model architectures to show U-Net successfully extracts stable features and improves performance compared to other models; see ***Section 5***, ***Appendix F***, and ***Table 5***.
- Measured model performance without stochasticity to show the effects of stochasticity in the next brick position selection improves model performance, especially in shape completion; see ***Appendix G*** and ***Table 6***.
- Showed that our model can assemble shapes without any squeezing-in bricks with slight modification on our model architecture; see ***Appendix H*** and ***Figure 6***.
- Analyzed the diversity of generated shapes from our model using a precision-recall graph; see ***Appendix I*** and ***Figure 7***.
- Tested brick assembly while considering intermediate stability using a physics simulator, PyBullet; see ***Appendix J***, ***Figure 8***, and ***Table 7***.
- Presented more additional qualitative results of our model; see ***Appendix K*** and ***Figure 9***.
- Revised a description of Potential Applications; see ***Section 6***.
- Updated an abstract by reflecting revisions; see ***Abstract***.


We will answer more detailed questions in the respective replies.

---

### Author Response · Authors · 2022-11-28
**Regarding further questions or concerns**

We deeply appreciate your comments again.

If you have further questions or concerns after reading our responses, please let us know.

---

### Decision · Program_Chairs · 2023-01-20

**Decision:**

Reject

**Justification For Why Not Higher Score:**

The paper is poorly written and the case for this method is weakened by other methods that work well on this problem.

**Justification For Why Not Lower Score:**

N/A

**Metareview: Summary, Strengths And Weaknesses:**

This paper could potentially be interesting, however it is poorly written. Detailed explanations of certain parts are OK, but the overall structure is confusing and shouldn't be published in the current form. To be specific, even the problem formulation takes a while to piece together. The basic problem and proposed solution should be outlined in the beginning and be clear. For example: 1. What is the problem space (e.g. 3d locations of 2x4 and 2x2 blocks or something else)?, 2. What is the objective (e.g. here is a 3d structure specified as such and such, find a sequence of brick locations that would assemble it or something else), 3. What is the objective optimised (e.g. overlap with target structure, sampled from a dataset or something else), 4. What is the dataset (structures, sequences?, made of lego or just generic shapes) and how do you use it. 5. What is the basic idea of your solution (e.g. RL on the objective function (no) or something else...).
As as second point, as reviewers pointed out, there are methods to solve this problem. Depending on who you ask, proposing the method in this paper might or might not be of value. In my opinion, if the method is interesting and general, it can still be of value even if solved by other means, but finding a problem where other methods don't work well would make a stronger case for the method.